

# Moving to 3D: relationships between coral planar area, surface area and volume

Jenny E. House[1], Viviana Brambilla[1], Luc M. Bidaut[2,3], Alec P. Christie[1], Oscar Pizarro[4], Joshua S. Madin[5,6] and Maria Dornelas[1]

[1] Center for Biological Diversity, Scottish Oceans Institute, School of Biology, University of St Andrews, St Andrews, United Kingdom
[2] Clinical Research Imaging Facility, University of Dundee, Dundee, United Kingdom
[3] College of Science, University of Lincoln, Lincoln, United Kingdom
[4] Australian Centre for Field Robotics, University of Sydney, Sydney, NSW, Australia
[5] Hawai'i Institute of Marine Biology, University of Hawai'i at Mānoa, Kaneohe, HI, USA
[6] Department of Biological Sciences, Macquarie University, Sydney, NSW, Australia

## ABSTRACT

Coral reefs are a valuable and vulnerable marine ecosystem. The structure of coral reefs influences their health and ability to fulfill ecosystem functions and services. However, monitoring reef corals largely relies on 1D or 2D estimates of coral cover and abundance that overlook change in ecologically significant aspects of the reefs because they do not incorporate vertical or volumetric information. This study explores the relationship between 2D and 3D metrics of coral size. We show that surface area and volume scale consistently with planar area, albeit with morphotype specific conversion parameters. We use a photogrammetric approach using open-source software to estimate the ability of photogrammetry to provide measurement estimates of corals in 3D. Technological developments have made photogrammetry a valid and practical technique for studying coral reefs. We anticipate that these techniques for moving coral research from 2D into 3D will facilitate answering ecological questions by incorporating the 3rd dimension into monitoring.

## INTRODUCTION

Coral reefs are one of the most diverse and more highly threatened ecosystems on the planet. Monitoring how corals respond to the vast array of threats and disturbances that they face (*Hoegh-Guldberg, 1999*; *Hughes et al., 2003*) is a crucial part of management and conservation. The challenge is understanding how best to quantify change in the corals themselves, and the wide range of ecosystem goods, functions and services which they provide (*Moberg & Folke, 1999*) to design effective monitoring programs (*Balmford, Green & Jenkins, 2003*).

Surface area and volume are 3D metrics particularly relevant for estimating the ecosystem services and functions performed by corals. Specifically, these two variables are critical for corals' reef building capability, which modulates many coral reef ecosystem services (*Moberg*

Corresponding author
Maria Dornelas,
maadd@st-andrews.ac.uk

& *Folke, 1999*), such as effective coastal defences (*Ferrario et al., 2014*) and biodiversity support (*Graham et al., 2006*). As such, coral volume, which is related to metrics such as biomass, growth rate and production of carbonate (*Cocito et al., 2003*), is a trait of primary interest for monitoring purposes. The importance of quantifying the reef in 3D also relates to the overall structure of the reef. In fact, the loss of this complexity is a major consequence of disturbance that leads to the degradation of biogenic habitats (*Airoldi, Balata & Beck, 2008*). While structural complexity can be maintained also by dead corals in the short term, many other ecologically significant functions, such as water filtering capability and productivity, are related to coral living surface area (*Cocito et al., 2003*). Microscale rugosity of colony surfaces facilitates larval recruitment (*Hata et al., 2017*), the continuous deposition of calcium carbonate ensures stability to the colonies structures and the predominance of living surface area in the colonies predicts abundance and survivorship of associated fauna (*Noonan, Jones & Pratchett, 2012*).

The proportion of live coral cover on a reef is probably the most widely used metric of reef health (*Leujak & Ormond, 2007*). A variety of techniques are used for estimating coral cover (*Loya, 1972*; *Hill & Wilkinson, 2004*; *Leujak & Ormond, 2007*; *Vroom, 2011*), most of which focus on 2D (planar) measurements of colony size or coral cover (*Gardner et al., 2003*; *Bruno & Selig, 2007*; *Sweatman, Delean & Syms, 2011*). The ubiquity of 2D representations of coral reefs enables standardization between and within different monitoring programs, allows them to be carried out on a range of spatial scales, and facilitates the fast collection of estimates of abundance and cover (*Shuman & Ambrose, 2003*; *Hill & Wilkinson, 2004*; *Booth et al., 2008*). However, there is increasing recognition of the need to develop better techniques for measuring coral colonies and reefs in 3D to account for different morphologies and complexity of coral colonies (*Burns et al., 2015a*; *Burns et al., 2015b*; *Goatley & Bellwood, 2011*; *Courtney et al., 2007*). For instance, overlooking the vertical aspect of coral reefs results in an inability to fully assess their structural complexity and measure ecologically significant changes (*Goatley & Bellwood, 2011*). In fact, coral morphotypes (also known as "growth forms") differ in their demographic rates and play distinct roles in the ecosystem. For example, morphotypes differ in their response to disturbance (*Madin & Connolly, 2006*) in their mortality schedule (*Madin et al., 2014*), fecundity (*Álvarez-Noriega et al., 2016*) and growth rates (*Dornelas et al., 2017*), and affect habitat complexity at different scales (*Richardson, Graham & Hoey, 2017*). Moreover, changes in the relative abundance of different morphotypes of corals may influence the provision of ecosystem services and biodiversity (*Alvarez-Filip et al., 2011*; *Burns et al., 2015b*). Using 3D approaches to better understand the structure and function of different coral morphotypes, as well as their vulnerability to disturbance, is an important step towards elucidating the goods and services that reefs provide.

In comparison to 2D techniques, methods that collect 3D data in the field are costly, time consuming and difficult to carry out (*Laforsch et al., 2008*; *Naumann et al., 2009*; *Goatley & Bellwood, 2011*), in addition to often being invasive or imprecise (*Naumann et al., 2009*). To overcome these problems, Structure-from-Motion (SfM) photogrammetric approaches became very useful in underwater settings since they permit the construction of three-dimensional digital models of objects beginning with a sequence of pictures of
the object itself. These techniques no longer require specification of known 3D locations prior to calculating camera positions (*Westoby et al., 2012*) and the ability to automatically match corresponding points across images. The precision and accuracy of this method has been demonstrated at different scales (*Figueira et al., 2015*; *Gutierrez-Heredia et al., 2016*; *Storlazzi et al., 2016*).

A growing number of studies present 3D reconstructions of coral reefs using SfM through proprietary software (*Burns et al., 2015a*; *Burns et al., 2015b*; *Burns et al., 2016*; *Raoult et al., 2016*; *Raoult et al., 2017*) or their own algorithms (*Friedman et al., 2012*; *Ferrari et al., 2016*; *Pizarro et al., 2017*). However, open access software is now available and being used among coral reef scientists (*Lavy et al., 2015*; *Figueira et al., 2015*; *Gutierrez-Heredia et al., 2016*; *Agudo-Adriani et al., 2016*), making SfM more accessible to a broad community. Therefore, calls for a switch towards 3D monitoring programs are gaining traction (*Raoult et al., 2016*; *Pizarro et al., 2017*). Assessments of measurement error associated to this technique are a fundamental part of the transition towards monitoring reefs in 3D (*Bryson et al., 2017*). Moreover, the consequences of this shift for loss of comparability with past monitoring, which largely involves planar imagery need to be assessed.

This paper addresses two aspects of moving from measuring corals in 2D to 3D. First, we ask whether we can predict 3D metrics of coral size from 2D metrics. We hypothesize that coral morphotypes differ in their scaling relationships between 2D and 3D metrics. The second aim of our study is to measure corals in 3D directly. We determine whether SfM provides accurate estimates of the surface area and volume of coral skeletons, and ask whether there are biases in this technique associated to different morphotypes. If the first assumption is met and SfM is demonstrated reliable, then a methodological shift in monitoring towards measuring traits in the three dimensions is possible, without losing the possibility to compare the results with previous data. The methodology developed allows including 3D metrics into coral reef monitoring, improving how we quantify change in coral reefs.

## METHODS

In order to capture 2D and 3D data, we used three methods for measuring coral skeletons, as outlined in Fig. 1. First, we measured colony planar total surface area (PL TSA) from birds-eye-view photographs of the colonies with a scale. Computed tomography (CT) scans and photogrammetry (PH) were used on the same specimens to produce information about 3D metrics, namely colony total surface area and volume (hereafter abbreviated to CT TSA, CT Vol, PH TSA and PH Vol respectively). In order to explore the most biologically useful information, the surface area of the colony that had been covered in corallites was also measured. This ''live'' surface area was produced from the results of all three methods; planar photography (PL LSA), photogrammetry (PH LSA) and CT scanning (CT LSA). Due to its high resolution, accuracy and inherent 3D nature, the data collected using CT scans was used as a baseline (*Veal et al., 2010*) with which to compare the other two methods. Using this suite of techniques enabled us to examine the relationship between 2D and 3D metrics, as well as address some of the difficulties with collecting 3D data.

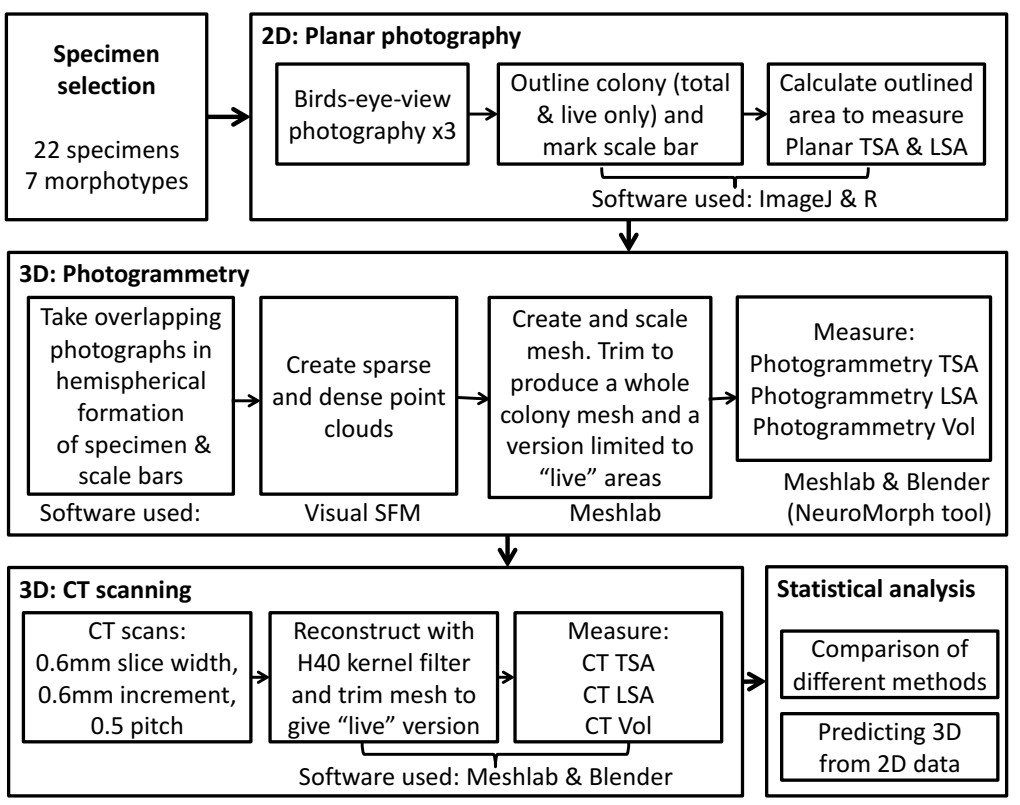

**Figure 1** The complete process used to measure TSA, LSA and volume in 2D and 3D for each specimen, including the measurement techniques and software used.

## Specimen selection

We selected coral skeletons from the collection at the Bell Pettigrew Museum, University of St Andrews with replicate specimens across different morphotypes and sizes. Each specimen was identified to species, and their morphotype was classified as branching, encrusting or massive. The resulting selection of coral skeletons includes 22 specimens described in Table S2.

## Photography and planar surface measurement

Coral specimens were photographed in air from above with a 10 cm × 10 cm chessboard-style calibration pattern using a digital camera (Nikon D40, 18–55 mm lens, Tokyo, Japan) as seen in Fig. 2A. The specimens were positioned on the plane in such a way as to replicate their natural orientation on the reef as much as possible. Each coral skeleton was photographed three times to account for and minimize the effect of measurement error. The specimens were repositioned for each photograph so as to minimize bias resulting from a particular position or camera angle.

All of the photos were then processed using the image analysis software ImageJ (*Rasband, 2014*). For each step the image was zoomed in in order to have the entire colony and scale completely in view and as big as possible. A graphics tablet (medium Intuos, Wacom, Kazo,

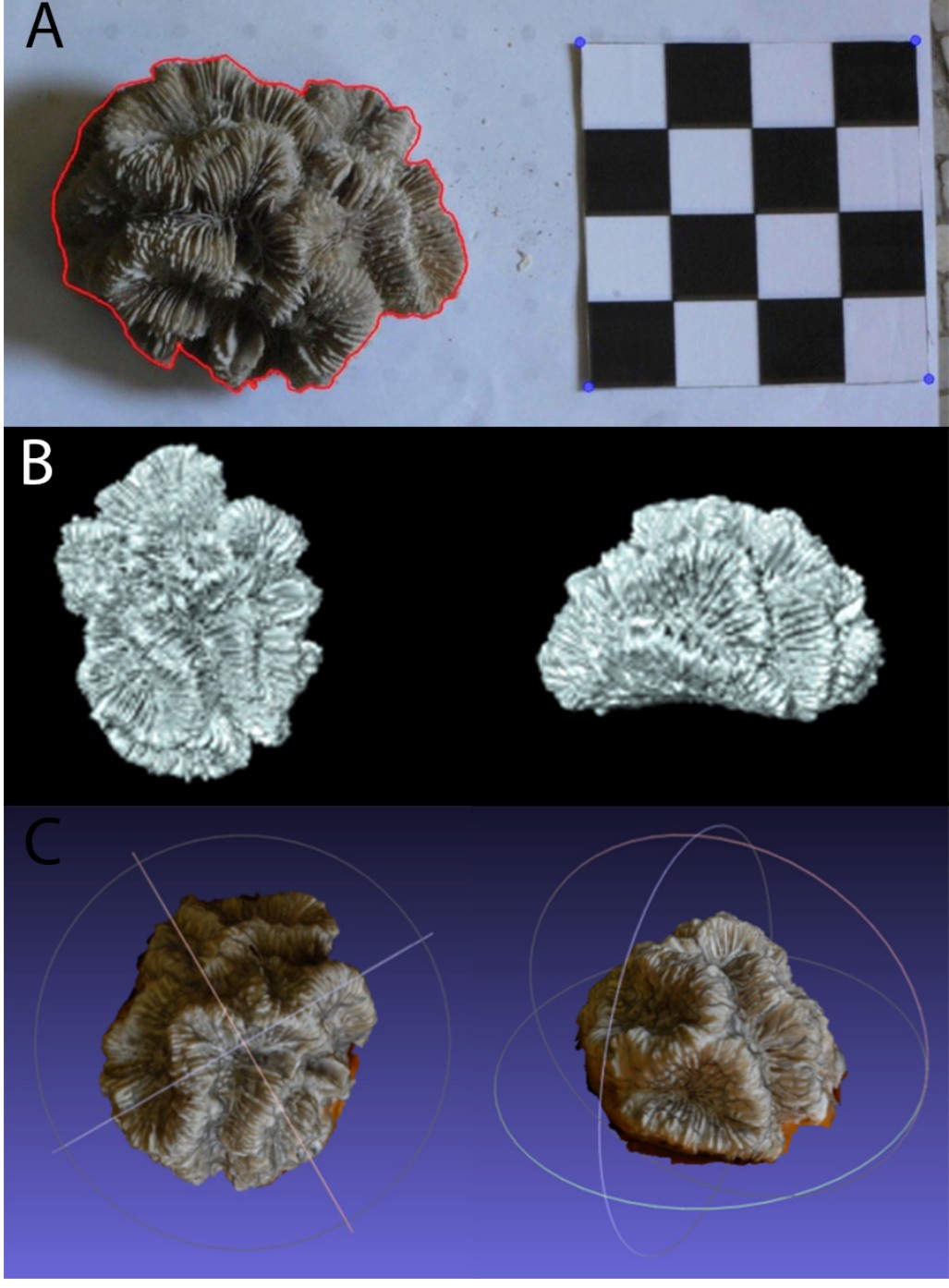

**Figure 2** Example of (A) planar photography of a coral colony having been outlined and scaled using ImageJ and R, (B) the surface generated using CT scanning, and (C) the equivalent surface generated using SfM photogrammetry.

Japan) was used to draw the outline of the whole coral colony and the areas that consisted of corallites. These contours were saved as a series of *XY* coordinates. The corners of the calibration pattern were also marked and saved as coordinates, in order to convert the pixel measurements into length (cm). After the necessary information had been extracted from the images and converted into *XY* coordinates, R (*R Core Team, 2013*) was used to calculate PL TSA and PL LSA from the relevant outlines (in square centimetres), using methodology and code from *Madin et al. (2014)*.

## Structure from motion

Photographs were taken using the same digital camera and a static off-camera flash set-up as for the planar photography. The specimens were placed on a table with four 10 cm scale bars positioned in a square on the surface around them. Photographs were taken with the camera positioned at various locations on a virtual hemispherical dome above the specimen, as illustrated in Fig. 3. This created a hemisphere-like spread of images of the specimen from various viewpoints. Significant overlap between images is needed in order to automatically identify shared points that can then be reconstructed as 3D coordinates. The number of views varied from 39 to 164 based on the size and complexity of the specimen. Specimens with occluding structures require the highest number of photographs in order to produce the necessary coverage. Distortions were not corrected.

Digital model construction was done on an Intel Quad Core 3.40 GHz desktop computer with 16 GB RAM under Windows 7 Professional. The open-source software package Visual SFM (*Wu, 2007*; *Wu, 2011*; *Wu, Frahm & Pollefeys, 2011*) was used to create a point mesh from the overlapping images by determining camera positions and generating a sparse point cloud. This was then followed by dense reconstruction using an additional package for Clustering views for Multi-View Stereo (CMVS) and Patch-based Multi-View Stereo (PMVS v2) (*Furukawa et al., 2010*; *Furukawa & Ponce, 2010*).

The dense point cloud was then imported into MeshLab (*Cignoni, Corsini & Ranzuglia, 2008*) and spurious points were removed. A surface layer was created from the point mesh using Poisson Surface Reconstruction. The scale bars were used to determine the coefficient needed to convert the mesh from pixels to absolute units, in this case millimetres. The model was then trimmed to remove the table and non-coral objects, as shown in Fig. 2C. The volume and surface area for these meshes were calculated using Blender (http://www.blender.org) with the NeuroMorph plug-in (*Jorstad et al., 2014*), thus producing PH TSA and PH Vol. To include PH LSAs, that are ecologically more meaningful than the specimen's entire surface area, the surface portions corresponding to not living corallites were selected and removed in MeshLab. To reduce the influence of any measurement errors, three models were produced for each specimen using different sets of images.

## Computed tomography and 3D surface measurement

The coral specimens were scanned in air using a medical CT scanner, Siemens Biograph mCT-128. The protocol was based upon that of *Naumann et al. (2009)*. The images were acquired at 0.6 mm slice width, 0.6 mm increments and 0.5 pitch. X-ray tube voltage was 120 kV with effective mAs of 341 (automatically varied) and a field of view that was

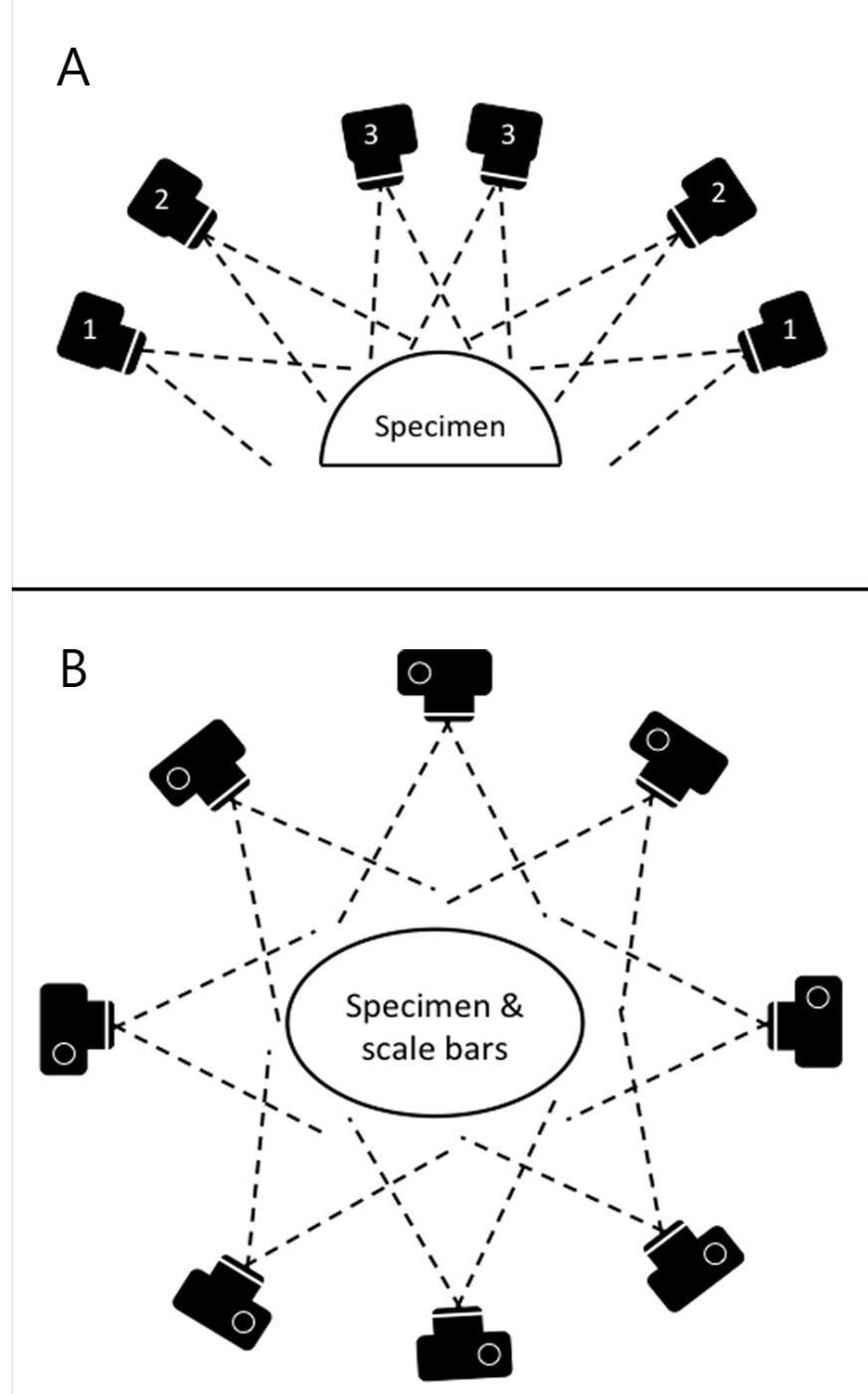

**Figure 3 Schematic of camera positions used to produce images for SfM photogrammetry.** In order to ensure appropriate coverage from multiple angles, we photographed the skeletons along three rings of at different heights (A) ensuring images taken were overlapping (B).

adapted to the size of each specimen. Three back-projection reconstructions were then produced for each colony from the spiral mode acquisition dataset, with sharp, medium and smooth kernel filters (H30, H40 & H50). Of these, the H40 reconstruction was selected for subsequent calculations because it gave the best compromise between high spatial resolution and low image noise. Using the corresponding 3D reconstructions of the coral colonies (example shown in Fig. 2B), measurements of CT TSA and CT Vol were generated in square and cubic millimeters, respectively. As with the meshes produced through SfM, Meshlab was used to trim away areas without corallites, and the CT LSA was then measured in Blender through the NeuroMorph toolset. Examples of CT and PH models are included in Fig. 2 and Fig. S1.

### Statistical analysis

CT metrics of size are used as response variables in our models since scanning can detect surface rugosity at a scale as small as 1,000 $\mu m^2$ (*Veal et al., 2010*) and provide the most accurate estimates of corals 3D features (*Laforsch et al., 2008*; *Naumann et al., 2009*; *Veal et al., 2010*). To address the first aim of testing whether 3D metrics can be inferred from 2D metrics of size, we fitted Ordinary Least Squares linear models predicting CT TSA and CT Vol from PL TSA and morphotype, and CT LSA from PL LSA and morphotype. Models with and without morphotype were compared using the Akaike Information Criterion (AIC) to assess whether differences in scaling among morphotypes affect the compromise between goodness of fit and model complexity. In addition, Adjusted $R^2$'s were used to assess the predictive ability of the different models.

Our second aim was to assess the ability of photogrammetry to estimate 3D metrics of coral size. As per the previous aim, we fitted Ordinary Least Squares linear models predicting CT TSA, CT LSA, and CT Vol from PH TSA, PH LSA, or PH Vol and morphotype. We performed model selection as above to investigate morphotype associated bias in the estimates. Finally, we compared Adjusted $R^2$'s of these models with those of a model with slope 1 and intercept 0.

These models used single measurements for CT TSA and CT Vol, but mean values were used for each specimen's PL TSA/LSA, PH TSA/LSA and PH Vol. All variables were log transformed to improve symmetry in the distribution of the residuals and to linearize the relationship between area ($mm^2$) and volume ($mm^3$). Statistical analysis was carried out in R (*R Core Team, 2013*).

## RESULTS

Colony planar area (PL TSA and PL LSA) can be used to infer accurate estimates of surface area and volume of the CT scanner models (CT TSA, CT LSA and CT Vol). As expected, 3D surface area is higher than 2D area, however the former scales tightly with the latter (Figs. 4A and 4B). Also, 3D volume is lower than the volume of a cube with a similar area, but again the scaling relationship is remarkably tight (Fig. 4C). The proportion of variance explained by the best model for each of these variables ranges between 0.81 and 0.90 (Table 1). Model selection suggests that morphotypes differ in their scaling relationship only for CT LSA (Fig. 4B, Table 2). For both CT TSA and CT Vol the slope in the best

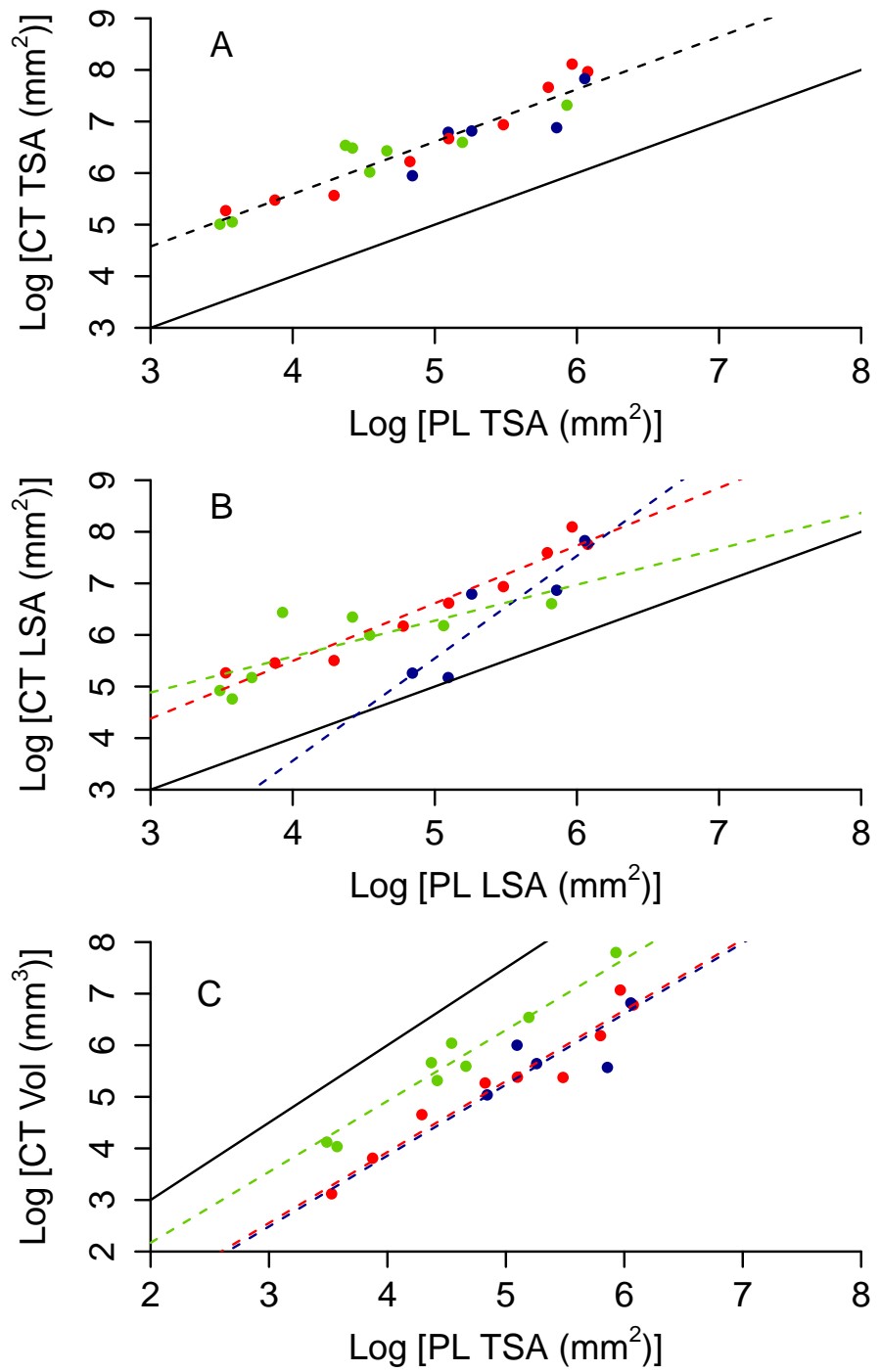

**Figure 4  3D metrics of size as a function of 2D metrics.** Red circles represent branching colonies, blue encrusting and green massive. The solid lines represent a model where 3D metric is equal to the 2D metric (A, B) or the relationship predict for a cube (C). Dashed lines represent predictions for the best model, with different colours for different morphotypes as per the symbols when morphotypes differ in parameter estimates.

Table 1 **Predictive accuracy of planar total or live surface area (PL TSA/LSA) when used alone and with morphotype to estimate CT TSA, CT LSA, CT Vol, respectively.** Adjusted $R^2$, $p$-value and Akaike's Information Criterion (AIC) are given to two significant figures.

| Response | Predictor (s) | Adjusted $R^2$ | $p$ | AIC |
|---|---|---|---|---|
| | PL TSA | 0.88 | $5.40 \times 10^{-11}$ | 15.03 |
| CT TSA | PL TSA + morphotype | 0.88 | $2.55 \times 10^{-8}$ | 16.01 |
| | PL TSA * morphotype | 0.88 | $7.22 \times 10^{-8}$ | 18.07 |
| | PL LSA | 0.70 | $3.65 \times 10^{-7}$ | 39.67 |
| CT LSA | PL LSA + morphotype | 0.745 | $3.55 \times 10^{-06}$ | 37.86 |
| | PL LSA * morphotype | 0.81 | $3.31 \times 10^{-06}$ | 32.88 |
| | PL TSA | 0.73 | $2.42 \times 10^{-07}$ | 42.85 |
| CT Vol | PL TSA + morphotype | 0.90 | $9.76 \times 10^{-10}$ | 23.14 |
| | PL TSA * morphotype | 0.90 | $1.79 \times 10^{-8}$ | 23.67 |

Table 2 **Parameter estimates for best models to predict CT TSA, CT LSA and CT Vol from PL TSA or LSA to for coral colonies of a range of morphotypes.** All variables in the regression models were log transformed hence a general predictive function is $C = e^{\alpha + \beta \ln(P)}$, where $C$ is CT TSA, CT LSA or CT Vol and $P$ is PL TSA, or PL LSA as per Fig. 1.

| Response | Morphotype | $\alpha$ (CI) | B (CI) |
|---|---|---|---|
| CT TSA | All | 1.528 (0.692 to 2.365) | 1.016 (0.849 to 1.184) |
| | Branching | 1.024 ($-0.749$ to 2.797) | 1.118 (0.768 to 1.468) |
| CT LSA | Encrusting | $-4.387$ ($-10.597$ to $-0.225$) | 1.987 ($-0.093$ to 1.830) |
| | Massive | 2.796 ($-0.812$ to 4.355) | 0.696 ($-0.975$ to 0.132) |
| | Branching | $-1.570$ ($-2.671$ to $-0.469$) | |
| CT Vol | Encrusting | $-1.638$ ($-0.501$ to 0.364) | 1.375 (1.160 to 1.589) |
| | Massive | $-0.579$ (0.610 to 1.373) | |

model is constant across morphotypes, although for CT Vol morphotypes differ in their intercept (Fig. 4, Table 2).

Photogrammetry provides fairly accurate estimates of the surface area and volume of coral skeletons: $R^2$ of best fit models range between 0.70 and 0.97 (Table 3 and Figs. S2 and S3). However, paired $t$-tests showed that the techniques for measuring 3D information, CT scanning and photogrammetry, produced significantly different measurements from each other for specimen volume ($t = -2.549$, $df = 21$, $p = 0.019$), TSA ($t = 2.91$, $df = 21$, $p = 0.008$) and LSA ($t = 3.518$, $df = 21$, $p = 0.002$). Photogrammetry generally underestimated TSA and overestimated volume (Fig. 5). Both photogrammetry and planar photography were less accurate at predicting CT LSA than CT TSA. Model selection does not reveal bias associated to morphotype for TSA and LSA, as the best model has constant scaling across morphotypes (Figs. 5A and 5B, Table 3). In contrast, the best model for Vol does include different slopes and intercepts for different morphotypes, as for massive colonies PH Vol is virtually identical to CT Vol, but for both encrusting and branching colonies the PH Vol increasingly overestimates CT Vol as colony sizes increase (Fig. 5C).

**Table 3 Predictive accuracy of Photogrammetry total and live surface area, and volume (PH TSA, PH LSA, PH Vol, respectively) when used alone and with morphotype to estimate total and live surface area and volume according to CT scanning (CT TSA, CT LSA, CT Vol, respectively).** Adjusted $R^2$, $p$ value and Akaike's Information Criterion (AIC) are given to three significant figures.

| Response | Predictor(s) | Adjusted $R^2$ | $p$ | AIC |
|---|---|---|---|---|
| | PH TSA | 0.876 | $9.75 \times 10^{-11}$ | 16.319 |
| CT TSA | PH TSA + morphotype | 0.875 | $5.92 \times 10^{-9}$ | 18.074 |
| | PH TSA * morphotype | 0.868 | $1.84 \times 10^{-7}$ | 20.686 |
| | PH LSA | 0.702 | $3.64 \times 10^{-7}$ | 39.601 |
| CT LSA | PH LSA + morphotype | 0.692 | $3.55 \times 10^{-06}$ | 41.983 |
| | PH LSA * morphotype | 0.690 | $3.31 \times 10^{-06}$ | 43.560 |
| | PH Vol | 0.955 | $1.02 \times 10^{-06}$ | 3.271 |
| CT Vol | PH Vol + morphotype | 0.973 | $2.52 \times 10^{-10}$ | −6.432 |
| | PH Vol * morphotype | 0.976 | $6.45 \times 10^{-9}$ | −7.847 |

# DISCUSSION

We have improved our understanding of the relationship between 2D and 3D metrics of coral colonies size and outlined an approach for converting between the two. For the size range investigated, our results support the hypothesis that 3D metrics of size scale consistently with planar area. Moreover, we demonstrated the potential for SfM to predict surface area and volume of the CT scanner models (CT TSA/LSA and CT Vol). Together, our results suggest that: (i) 2D data can be converted into more ecologically meaningful 3D metrics, such as colony surface area and volume, when combined with information about colony morphotypes, and (ii) that a shift towards 3D indicators in monitoring programs is possible, without losing comparability in the process.

The measurements collected using SfM models were found to be significantly different from the results of the CT scans, but were nevertheless excellent predictors when combined with information about the morphotype of the colony. The differences observed are linked to the different resolutions of the two methods (much higher for CT scans, see Fig. S1). Lower resolution 3D SfM models cause both the underestimation in surface area and the overestimation in volume. Our study adds to growing evidence that the previously prohibitive aspects of underwater photogrammetry are being overcome by technological improvements (*Burns et al., 2015a*; *Falkingham, 2012*). Not only the possibility of applying this technique using open-source software opens it up to a wider audience, but the costs of specific all-in-one SfM software is decreasing and allowing greater control upon reconstruction parameter and resolution. The application of photogrammetry to measuring reef topography (*Burns et al., 2015b*) combined with our detailed modelling of individual coral colonies illustrates the wide range of potential applications this technique can have in monitoring and studying coral reefs and their ecology at different scales.

Quantifying size in 3D rather than 2D is time costly in both the field and the lab. We found that SfM photogrammetry was easier to carry out when dealing with less complicated morphotypes, which required less processing time and fewer photographs. SfM is particularly effective for colonies with simpler structures and few occlusions, and
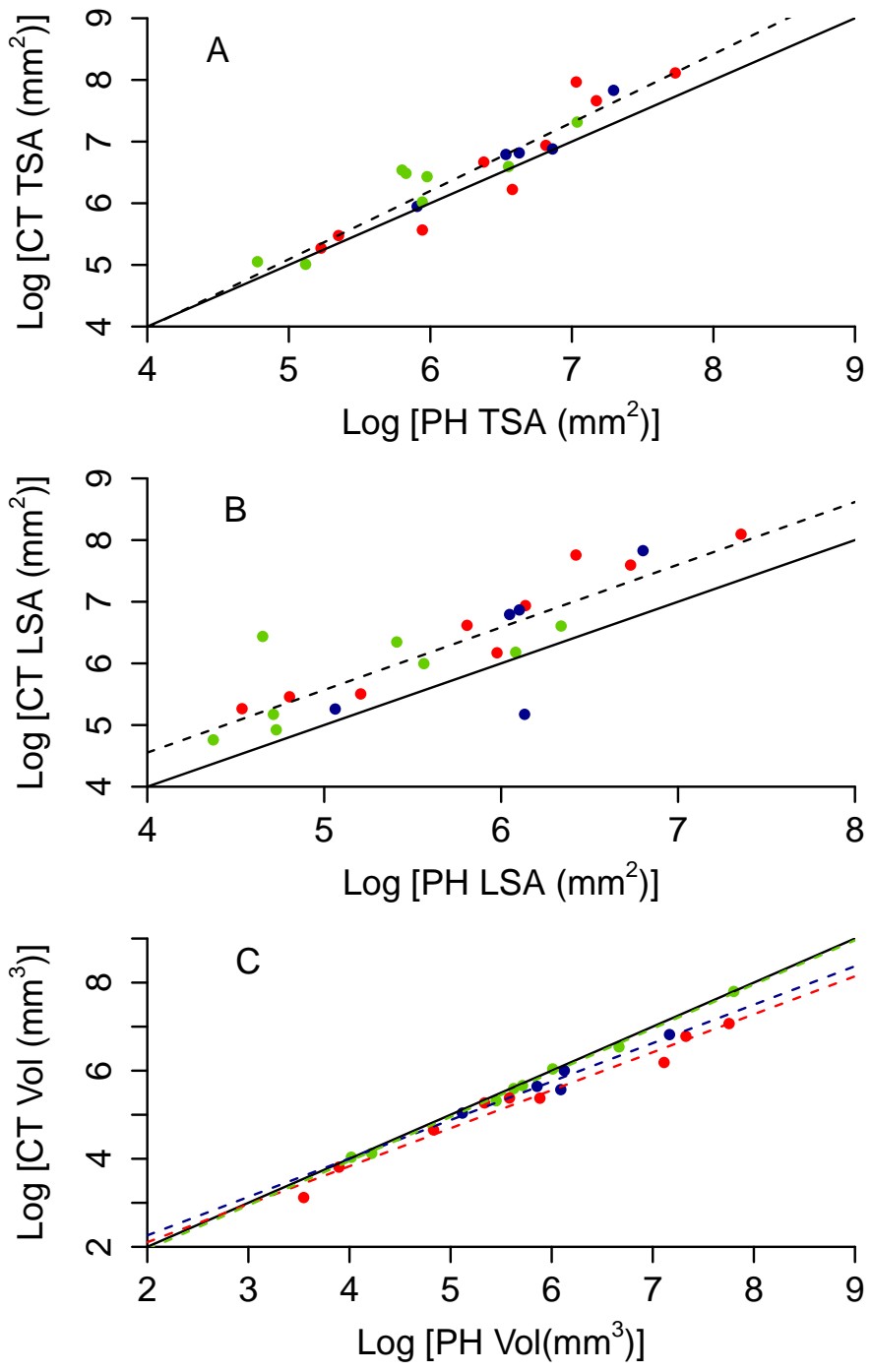

**Figure 5** **Relationship between CT and PH estimates of colony size (A total surface area, B live surface area and C volume).** Red circles represent branching colonies, blue encrusting and green massive. The solid lines represent a model where the two metrics are identical. Dashed lines represent predictions for the best model, with different colours for different morphotypes as per the symbols when morphotypes differ in parameter estimates.

it has been suggested that it could be a valuable technique in areas with a high prevalence of hemispherical colonies, such as the Caribbean (*Courtney et al., 2007*). In contrast, calculating PL TSA/LSA took less time because it required fewer photographs and less image processing. Although more complicated morphotypes still required more processing than simple colony shapes, the difference in time and effort was negligible compared to when using the photogrammetric approach. The labour-intensive nature of measuring corals in 3D, despite recent technological developments, does suggest that the option of converting 2D measurements into 3D metrics may provide a useful alternative in cases where conducting monitoring in 3D is not feasible due to the time or costs involved. Additionally, SfM software improvements are making this approach faster, more efficient and streamlined.

The most beneficial outcome from this study is that previously collected coral cover data may converted into 3D metrics if morphotype was captured during collection (e.g., as categorical data in line intercept transects or by re-analyzing video transects). This can help shifting towards 3D without losing the possibility of comparisons with past measurements, overcoming in a smooth way the "methodological inertia" that characterizes monitoring programs (*Goatley & Bellwood, 2011*). We have produced empirical formulae that combine PL TSA/LSA and morphotype categories to predict colony TSA/LSA or volume. Importantly, the predictive power of these conversion models is similar to the estimates obtained through SfM. Our results indicate the importance of recording the morphotype of a colony when conducting monitoring, as this trait determines the relationship between some of the 2D and 3D metrics. Increasing the number of specimens for each morphotype and widening the size spectra would further improve these formulae, and it would be valuable also to expand them to additional morphotypes in the future.

Morphotype categories are not always clear-cut and the variability within groups supports the need to move from discrete classifications of morphotypes towards individual level continuous traits that measure colony shape. Moreover, our work suggests that surface area and volume, as well as the ratios between these variables and planar area, are potential candidates as useful traits. This shift in focus would also address the fact that corals can exhibit a high degree of morphological plasticity within species (*Todd, 2008*), with colonies of the same species fulfilling different categories of morphotype. This level of plasticity suggests that when our equations are used in the future they should be applied based on the morphotype observed in the field, rather than one that is based on species identification and applied *post hoc*.

Improved understanding of the relationship between 2D and 3D parameters for different morphotypes should contribute towards our grasp of the ecological role of different coral morphotypes. We already know that morphotypes respond differently to disturbance (*Madin & Connolly, 2006*) and play different ecological roles (*Alvarez-Filip et al., 2011*). It has also been suggested that examining the ratio of different coral morphotypes on reefs can give insight into reef health (*Edinger & Risk, 2000*). Our approach can provide a transition between traditional methods and accurate 3D modelling, which will improve our understanding of the contribution of different morphotypes to the services and functions provided by coral reefs. In addition to applying our findings to future research, a significant

benefit of using the equations developed herein is that they can be applied to archived images and historical data sets. This will enable data comparisons over as long a timescale as possible, minimising the "shifting baseline" effect (*Knowlton & Jackson, 2008*).

## CONCLUSIONS

In conclusion, coral colony surface area and volume can be predicted effectively from two commonly collected variables: planar area (PL TSA) and morphotype. This quantitative development provides a stepping-stone that may enable better understanding and exploitation of historical data. Furthermore, SfM photogrammetry clearly contributes towards addressing the question of how best to measure corals because it is a widely accessible, non-invasive and cost effective method for making 3D measurements in-situ. This paper illustrates two specific areas for studying corals in ways that better capture changes amongst corals and the ecological processes associated with them. We hope that these approaches will eventually enable more accurate coral reef monitoring and conservation.

## ACKNOWLEDGEMENTS

We thank the curator and staff at the Bell Pettigrew Museum for allowing us use of museum specimens, and the Behaviour and Biodiversity group at University of St Andrews for feedback.

### Funding

This work was supported by the School of Biology, University of St Andrews, the Scottish Funding Council (MASTS grant reference HR09011) and the Templeton Foundation (grant #60501, 'Putting the Extended Evolutionary Synthesis to the Test'). The funders had no role in study design, data collection and analysis, decision to publish, or preparation of the manuscript.

### Grant Disclosures

The following grant information was disclosed by the authors:
School of Biology, University of St Andrews, the Scottish Funding Council: HR09011.
Templeton Foundation: #60501.

### Competing Interests

Maria Dornelas is an Academic Editor for PeerJ.

### Author Contributions

- Jenny E. House conceived and designed the experiments, performed the experiments, analyzed the data, wrote the paper, prepared figures and/or tables, reviewed drafts of the paper.

- Viviana Brambilla analyzed the data, wrote the paper, prepared figures and/or tables, reviewed drafts of the paper.
- Luc M. Bidaut performed the experiments, contributed reagents/materials/analysis tools, reviewed drafts of the paper.
- Alec P. Christie performed the experiments, reviewed drafts of the paper.
- Oscar Pizarro contributed reagents/materials/analysis tools, reviewed drafts of the paper.
- Joshua S. Madin conceived and designed the experiments, reviewed drafts of the paper.
- Maria Dornelas conceived and designed the experiments, analyzed the data, wrote the paper, reviewed drafts of the paper.

## Data Availability

The raw data and R script have been supplied as Supplementary Files.

## Supplemental Information

Supplemental information for this article can be found online at http://dx.doi.org/10.7717/peerj.4280#supplemental-information.

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
