# Peer review of "Moving to 3D: relationships between coral planar area, surface area and volume"

_PeerJ, doi:10.7717/peerj.4280_

## Round 0.1 · original submission · Major Revisions

Dear Dr. Dornelas,

We have received the reports from three reviewers on your manuscript “Moving to 3D: relationships between coral planar area, surface area and volume”. See below the comments provided by the reviewers, who performed a detailed work. I must inform you that, based on the advice received, your manuscript may be considered acceptable for publication in PeerJ after a major review is performed. Please, consider all the referees remarks in your revised manuscript and indicate in your rebuttal letter a point-by-point response to the referees.
With kind regards,
Ronaldo

·

Basic reporting

Overall this paper provides valuable research into the growing field of photogrammetry from SfM in marine environments. The methodology is sound, and the use of planar conversions could be a helpful tool for management purposes. However, this paper suffers from a lack of cited literature that relates to the field of photogrammetry from SfM in coral reefs. Please see below for comments:

The submission adheres to all PeerJ policies.

The article must be written in English using clear and unambiguous text and must conform to professional standards of courtesy and expression: In general this is true, though there are some minor errors in the way the introduction is structured.

Lines 7-8: making this statement at the end of the first paragraph is confused, it would fit better towards the end of your intro.

The article should include sufficient introduction and background to demonstrate how the work fits into the broader field of knowledge. Relevant prior literature should be appropriately referenced.

This is primarily where the article falls short. Numerous times throughout the introduction (primarily lines 61-81) false statements are made regarding the efficacy or use of photogrammetry/SFM. For example, lines 63-66 ignores recent published studies, e.g. Raoult et al., 2016, Ferrari et al., 2016, Figueira et al., 2015, Storlazzi et al., 2016, Lavy et al. 2015 that all use photogrammetry/sfm to measure 3D aspects of coral reefs. All these studies use low-cost imagery techniques (action camera or digital camera in dive housing) to produce metrics from corals/coral reefs. The paragraphs lines 61-81 need to be heavily revised to take into account the recent developments of the use of SfM for managing coral reefs.

The definition of photogrammetry lines 68-69 is inaccurate: photogrammetry does not include building models. This is what structure from motion does (hence why the authors use VisualSFM first and a model measuring tool second).

The statement lines 78-81 is false: there are numerous published studies that use open-access software or freeware to study coral colonies (i.e. Lavy et al. 2015, Gutierrez-Heredia et al. 2016, Figueira et al. 2015, Agudo-Adriani et al. 2016).

Lines 78-81: I would add a caveat to this sentence. While open-access software is available, it is generally less user friendly, less streamlined in processing (i.e. requires three separate types of software) and the computational requirements are often greater than off-the-shelf SfM specific payware (an issue that the authors appear to run into). Numerous studies have turned to Agisoft Photoscan for SfM processing (i.e. Raoult et al. 2016, Burns et al. 2016, Burns et al. 2015) as the most efficient all-in-one software (SFM to measurement), and some studies have turned to creating their own algorithms (i.e. Ferrari et al. 2016).

Experimental design

Lines 117-118: It is not made explicit whether corals were taken out of their natural environment (i.e. pictures taken in the air or underwater). This is important because immersed cameras in dive housing generally have higher indices of reflection, which can distort images. If a dive housing was used, provide the model. You also provide a lens (18-55mm) but do not state at what focal length the images were taken. This can have an impact on the accuracy of SfM models if it is not corrected appropriately.

Lines 125-127: What is the rationale for zooming in (I assume 'cropping' is the intended meaning)? Is it just to reduce processing time by having fewer points to align?

Figure 2: The bottom image with SfM models needs to be re-aligned, it looks messy (partial overlap). Include a) b) c) on the image itself rather than only in the caption.

Lines 159-163: It does not appear that any corrections for lens distortion were made. This can have an impact on the accuracy of SfM, and should at least be stated.

Citations are inconsistent and/or have errors throughout the manuscript. Please ensure you use a consistent referencing style and check for errors.

Line 159: Error in citation (should be Wu et al., 2011).

Line 171: Error in citation (should be Jorstad et al., 2014)

Line 171-173: This sentence is confusing and needs to be reworded. It would also fit better after the sentence lines 173-174.

Line 220: Should be "The model" rather than "model"

Linees 249-250: Put spaces between = signs. Include all the statistical information (df, statistical value)

Validity of the findings

The conclusions (lines 275-275) need to be worded more precisely. You found a planar relationship with 3D structure, however, it appears many of your corals were small in size, and as a result your planar area would be closely related to volume or TSA/LSA. If, however, you measured a very vertical colony (i.e. a highly branching species), a planar image would greatly underestimate TSA and volume. Perhaps underline that these models are valid for small colony sizes? Line 80 you talk about measuring corals on a large scale, yet you have not done so in this study. Be careful making broad statements like this.

Do not interchange photogrammetry and SfM throughout your text. Photogrammetry is the measurement of points within models, SFM is how those models are constructed. You should generally refer to 'SFM models' and 'measurements using photogrammetry'

Lines 286-289: You can reliably increase the resolution of SfM models by increasing the resolution of the camera, taking more pictures at various distances, or within the program itself (this is where VisualSFM falls short, because you have little control over resolution settings).

Paragraph line 295: You state there are issues with processing time. It would be very beneficial to provide the specifications of the computer used to construct the point clouds in the methods and to provide a range of processing time (if available). Again, VisualSFM does not use graphics processors in PCs that can more than halve the processing time. Again, do not use photogrammetry so liberally: you generally mean SfM in this paragraph.

Line 303-305: This statement needs greater consideration of the benefits of streamlined software designed for this task (you can get the same measurements in Agisoft Photoscan with 4 steps and a dozen or so human inputs).

Lines 304-307: It's hard to foresee a situation where the costs (financial? labour? time?) for SfM would be so great as to suggest planar extrapolation given the benefits of 3D measurements over planar ones. I think the benefit of this technique would primarily be for historical data that has already been retrieved.

line 308: again, photogrammetry is not a 3D technique, you mean SfM here

Lines 309-310: This is a crucial statement. The issues with SfM are generally from the camera end rather than the processing end

Line 313: Laser scanners are definitely becoming more financially viable, but they don't obtain information on living or dead coral. I'm also not sure what the benefit of stereo cameras would be in this application, since they generally are processed with SfM anyway.

Paragraph line 316: Yes, this entire paragraph is what you should focus on. I think it would be better placed just after the first paragraph in your discussion. It also has a bit of a nonsensical statement: if you are taking planar images of coral, presumably it is not necessary to record morphotype.

Paragraph line 327: it may be relevant to discuss classification of corals here, as recent projects focus on coralite structure for classification (i.e. Coral Finder) as opposed to traditional morphotypes.

Conclusions: Need to reword line 356, as the shift is already occurring (see citations provided)

Additional comments

Citations that are relevant and should be included in this paper:

Raoult, V., David, P.A., Dupont, S.F., Mathewson, C.P., O’Neill, S.J., Powell, N.N. and Williamson, J.E., 2016. GoPros™ as an underwater photogrammetry tool for citizen science. PeerJ, 4, p.e1960.

AGUDO-ADRIANI, E. A., CAPPELLETTO, J., CAVADA-BLANCO, F. & CROQUER, A. 2016. Colony geometry and structural complexity of the endangered species Acropora cervicornis partly explains the structure of their associated fish assemblage. PeerJ, 4, e1861.

BURNS, J., DELPARTE, D., GATES, R. & TAKABAYASHI, M. 2015a. Integrating structure-from-motion photogrammetry with geospatial software as a novel technique for quantifying 3D ecological characteristics of coral reefs. PeerJ, 3, e1077.

BURNS, J., DELPARTE, D., GATES, R. & TAKABAYASHI, M. 2015b. Utilizing underwater three-dimensional modeling to enhance ecological and biological studies of coral reefs. The International Archives of Photogrammetry, Remote Sensing and Spatial Information Sciences, 40, 61.

BURNS, J., DELPARTE, D., KAPONO, L., BELT, M., GATES, R. & TAKABAYASHI, M. 2016. Assessing the impact of acute disturbances on the structure and composition of a coral community using innovative 3D reconstruction techniques. Methods in Oceanography.

FERRARI, R., BRYSON, M., BRIDGE, T., HUSTACHE, J., WILLIAMS, S. B., BYRNE, M. & FIGUEIRA, W. 2016. Quantifying the response of structural complexity and community composition to environmental change in marine communities. Global change biology.

FIGUEIRA, W., FERRARI, R., WEATHERBY, E., PORTER, A., HAWES, S. & BYRNE, M. 2015. Accuracy and Precision of Habitat Structural Complexity Metrics Derived from Underwater Photogrammetry. Remote Sensing, 7, 16883-16900.

GUTIERREZ-HEREDIA, L., BENZONI, F., MURPHY, E. & REYNAUD, E. G. 2016. End to End Digitisation and Analysis of Three-Dimensional Coral Models, from Communities to Corallites. PLoS ONE, 11, e0149641.

LAVY, A., EYAL, G., NEAL, B., KEREN, R., LOYA, Y. & ILAN, M. 2015. A quick, easy and non‐intrusive method for underwater volume and surface area evaluation of benthic organisms by 3D computer modelling. Methods in Ecology and Evolution, 6, 521-531.

STORLAZZI, C. D., DARTNELL, P., HATCHER, G. A. & GIBBS, A. E. 2016. End of the chain? Rugosity and fine-scale bathymetry from existing underwater digital imagery using structure-from-motion (SfM) technology. Coral Reefs, 1-6.

Reviewer 2 ·

Basic reporting

No comments

Experimental design

83-84 / 196: Authors need to be more clear about exactly which metrics they are going to compare between 2D and 3D. Cover, abundance and size seem to be used interchangeably however they are not (necessarily) the same. I expect that the authors know this, but it could be more clear.

121: Three photographs each time the coral was repositioned or one for each of three orientations? How was this data used in the regressions? Was an average used or was each used as an independent data point? This will either lead to error in the predictor or pseudo-replication, both of which should be avoided in OLS linear models …

174: As written it is not clear why live surface area is more ecologically meaningful, especially in regards to the creation of 3D structure on the reef, which is present whether or not a calcium carbonate structure is covered with live coral tissue.

175: What is the magnitude of the error here? I am concerned that this was not reported, especially as this was used as a predictor variable (which should not have error). It would be very useful to know how variable the SfM reconstructions are.

194-201: This paragraph is confusing as written and the first sentence is misleading. Also, at least a sentence (or two) of why a CT scan can be used to validate photogrammetry from SfM should be included.

Validity of the findings

No Comments

Additional comments

I appreciate the approach and am confident in the quality and rigor of work presented here. This is a powerful and innovative tool that they are recommending which could lead to very important and needed changes in the ways that coral reefs are studied. However, the introduction does not provide a well formed biological argument to justify their approach. The discussion does a better job in this respect, but arguments could be elaborated. As they are recommending that these tools be used in monitoring, the authors should spend more time explaining the details of the meshing and calculation of area, which are non trivial and can result in fundamental differences in data types and quality. I would like to see this paper go in a more ecological direction or a more methodological direction (preferably the former) and recommend that the introduction and discussions be retooled to this effect. With these changes, I would be very happy to support this manuscript.

Line-by-line comments:
7: It is not clear how the methods used here represent a tool that can used to incorporate 3D metrics in monitoring. More specific biological / ecological argument explaining the utility of these metrics is needed.
21: The paragraph does not explain how changes in 3D metrics are more meaningful than changes in 2D metrics.
45-46: I’m having trouble with the logical basis of this argument (not that I necessarily disagree with it). A great deal of the paper is devoted to showing that 3D metrics can be estimated from 2D metrics. Is this a good justification for the incorporation of 3D approaches?
51-52: How do they modulate ecosystem services? How do they support reef assemblages? At several points in the manuscript examples are given in this manner without any elaboration. It would greatly improve the quality of the manuscript if there were specific biological and ecological explanations and not simply lists of examples.
83-84 / 196: Authors need to be more clear about exactly which metrics they are going to compare between 2D and 3D. Cover, abundance and size seem to be used interchangeably however they are not (necessarily) the same. I expect that the authors know this, but it could be more clear.
317-318: This argument is not well formed.
344: This paper presents data of single coral photogrammetry and planar imagery of single corals. It is not clear how these techniques can be used to measure numerical abundance or cover at the scale used by most monitoring programs.

·

Basic reporting

The submitted text is written in a professional English language, is clear and unambiguous as demanded by the guidelines. They contextualize the issue of obtaining geometric descriptors of corals, as they are used in most of the monitoring protocols of coral reefs, with key references. They argue in favor of shifting from 1D/2D approach to a 2D/3D approach, based on references that present modern techniques to obtain these parameters. There is only one concept that I think they need revising for the sake of precision, which is the concept of Photogrammetry. Photogrammetry is the science of obtaining reliable and accurate measurements of an object using photography. The use of multiple photographs to build a digital model is a specific activity of the science. So this is just a matter of rewrite the sentence. The structure of the paper conforms to PeerJ standard and discipline norm and is plainly clear. All figures are relevant and have good quality and descriptions but some lack labels (Figure 2). Raw data are supplied as Supplemental files but not for publication.

Experimental design

The paper presents an original research that is primarily Biological in scope, with a geometric approach as a tool for determining organismal surface and volume size, that is applicable to assess structural stability or biomass, for example. The two well-defined research questions are 1. Can one predict 3D metrics (volume) fro 2D metrics (area)? 2. Are photogrammectric techniques appropriate to estimate surface area and volume of coral skeletons of different morphotypes? These questions are relevant because one can increase accuracy of the metrics of corals that are useful in evaluating vulnerability and resilience of a reef system, based on coral morphology. The proposed photogrammetric technique was proposed as a cost effective tool to approach coral morphology. The number of coral replicates and data replication was appropriate to the accuracy and robustness of the statistical tests the authors applied. The methods are well described, with sufficient detail in general. However, when addressing the use of planar photographs, I found the description lacking details as to the calculation of the planar area on the photographs. They state “using methodology and code from Madin et al. (2014).” but I did not find any detailed calculation description or the R codes in the cited paper. There isn’t supplementary material in the publisher site either.

Validity of the findings

The results are robust, the statistics they used is sound and the conclusions, thus, robust either. They present the results in a straightforward manner. The figures illustrate well the relationship between the planar view descriptors and the CT derived descriptors, demonstrating the possibility of deriving the 3D metrics from the planar descriptors, if coral growth form is considered. They propose predictive functions for each descriptor per morphotype to be estimated from the planar descriptors. They also show that the photogrammetric technique is useful to calculate 2D and 3D descriptors, although still not as cost-effective as the planar photographs.

External reviews were received for this submission. These reviews were used by the Editor when they made their decision, and can be downloaded below.

---

## Round 0.2 · accepted · Accept

Dear Maria,

I am pleased to inform that your manuscript - Moving to 3D: relationships between coral planar area, surface area and volume - has been Accepted for publication in PeerJ.

Yours sincerely

Ronaldo